# The Role of the CX3CR1-CX3CL1 Axis in Respiratory Syncytial Virus Infection and the Triggered Immune Response

**DOI:** 10.3390/ijms25189800

**Published:** 2024-09-11

**Authors:** Selma Rivas-Fuentes, Alfonso Salgado-Aguayo, Teresa Santos-Mendoza, Edgar Sevilla-Reyes

**Affiliations:** 1Laboratory of Transcriptomics and Molecular Immunology, Instituto Nacional de Enfermedades Respiratorias Ismael Cosío Villegas, Mexico City 14080, Mexico; edgar.sevilla@iner.gob.mx; 2Laboratory of Research on Rheumatic Diseases, Instituto Nacional de Enfermedades Respiratorias Ismael Cosío Villegas, Mexico City 14080, Mexico; alfonso.salgado@iner.gob.mx

**Keywords:** RSV, CX3CL1, CX3CR1, immune response

## Abstract

Respiratory syncytial virus (RSV) is a common respiratory pathogen that causes respiratory illnesses, ranging from mild symptoms to severe lower respiratory tract infections in infants and older adults. This virus is responsible for one-third of pneumonia deaths in the pediatric population; however, there are currently only a few effective vaccines. A better understanding of the RSV–host relationship at the molecular level may lead to a more effective management of RSV-related symptoms. The fractalkine (CX3CL1) receptor (CX3CR1) is a co-receptor for RSV expressed by airway epithelial cells and diverse immune cells. RSV G protein binds to the CX3CR1 receptor via a highly conserved amino acid motif (CX3C motif), which is also present in CX3CL1. The CX3CL1-CX3CR1 axis is involved in the activation and infiltration of immune cells into the infected lung. The presence of the RSV G protein alters the natural functions of the CX3CR1-CX3CL1 axis and modifies the host’s immune response, an aspects that need to be considered in the development of an efficient vaccine and specific pharmacological treatment.

## 1. Generalities

Respiratory syncytial virus (RSV) is a major cause of respiratory tract infections in infants and young children; most children have been infected at least once with RSV by the age of two. In infants, RSV can cause severe lower respiratory tract infections, such as bronchiolitis, pneumonia, and other complications, leading to hospitalization and, in some cases, death [1]. RSV infections in infants can cause long-term pulmonary sequelae [2]. Although infrequent, extrapulmonary manifestations of RSV are severe and include heart failure, neurological involvement, and hepatitis [3,4,5].

In 2019, it was estimated that, globally, lower respiratory tract infections caused by RSV resulted in more than 3,000,000 hospital admissions and over 100,000 overall deaths in children aged 0–5 years [6]. RSV typically causes mild upper respiratory tract infections, but can cause otalgia [7] and acute respiratory distress in certain populations, such as older adults [8] and individuals with underlying health conditions [9]. Another feature of RSV in immunocompromised patients is that viral clearance is delayed, and the virus can remain in nasal secretions for a long time [1].

Despite the onset of the host’s immune response, RSV infection commonly elicits relatively short-lived protective immunity, resulting in recurrent seasonal RSV infections [10,11,12]. For young infants, the most effective prophylactic approach is passive immunization, naturally transferred, or by means of commercially available monoclonal antibodies Palivizumab and Nirsevimab, which inhibit RSV fusion with the host cell [13,14,15]. Nirsevimab has the advantage of an extended serum half-life, thus requiring only one administration per RSV season, and it is effective against hospitalization in over 70% of cases [15,16,17]. Currently, there are three FDA-approved vaccines against RSV: Arexvy, Abrysvo, and mRESVIA. These vaccines have been approved for use in older adults [18,19]. Furthermore, Abrysvo can be used to immunize pregnant women, thus conferring passive immunity to the newborn through the transplacental transfer of neutralizing antibodies [19]. mRESVIA is an mRNA-based vaccine and is the latest vaccine against RSV approved by the FDA, in May, 2024 [20].

RSV is a pleomorphic enveloped virus with three possible morphologies: spherical (ranging in diameter from 100 to 150 nm), asymmetric, and filamentous (with filament lengths from 0.5 to 12 μm) [21]. Its genome is a non-segmented, negative-sense, single-stranded RNA of 15.2 kilobases, comprising 10 ORFs and encoding 11 proteins: NS1, NS2, N, P, M, SH, G, F, M2.1, M2.2, and L [22]. RSV has two major antigenic groups, known as subgroups or subtypes A and B, although one usually predominates each season [23]. Based on the genetic diversity of the viral coded G protein, 13 genotypes have been proposed for RSV-A [24,25] and 37 for RSV-B [26], but due to conflicting data, multiple updates and reclassifications have been proposed. More recently, complete genome studies have led to groupings of these genotypes into lineages [27].

## 2. Infection

RSV primarily infects the ciliated epithelial cells lining most of the respiratory tract and type I alveolar pneumocytes, leading to cell death and disruption of the respiratory epithelium, often resulting in syncytia formation [28]. Some studies suggest that syncytia formation may vary based on the viral strain and host conditions [29,30].

The entry of RSV into host cells is initiated by the attachment of the viral glycoprotein G (referred to as G protein hereinafter) to host cell molecules [31] such as glycosaminoglycans (including heparan sulfate) and the transmembrane chemokine receptor CX3CR1 [32]. RSV protein F facilitates fusion and micropinocytosis by binding to nucleolin, epidermal growth factor receptor (EGFR), insulin-like growth factor-1 receptor (IGF1R), and intercellular adhesion molecule-1 (ICAM-1) [33,34]. The vaccines mentioned above induce an immune response to this protein. After fusion of the viral envelope with the host cell membrane, the viral ribonucleoprotein (RNP) complex (viral RNA stabilized by N and P proteins) enters the host cytoplasm [35]. The SH protein, functioning as a viroporin, may facilitate this process by altering membrane permeability, reducing apoptosis, and inhibiting TNF-α signaling [36].

Once inside the host cell, viral protein complexes form inclusion bodies in the cytoplasm to replicate RNA genomes and transcribe them into mRNA for protein synthesis by the host cell machinery [37]. 

Naked virions are assembled when the matrix protein (M) facilitates interactions between the new RNP complexes and viral proteins G, F, and SH [38,39]. The assembled virions bud from the host cell membrane and acquire their lipid envelopes from the host. [40]. Among the RSV proteins, the G protein plays a crucial role in infection because its CX3C motif allows the virus to interact with the chemokine receptor CX3CR1 [32,41]. As discussed in subsequent sections, this interaction is key to both viral infection and the host’s immune response [42]. 

## 3. Biological Aspects of CX3CL1-CX3CR1 

CX3CL1 is the only member of the CX3C chemokine subfamily [43]. It is characterized by being the only chemokine to have three amino acid residues between the two conserved cysteines all chemokines have. It is expressed in activated endothelial cells, epithelial cells, and neurons, among others [44,45].

Of the 50 known chemokines, only CXCL16 and CX3CL1 are synthesized as membrane-bound molecules [46]. CX3CL1 consists of a chemokine domain and a mucin stalk, which are both extracellular; it also has a transmembrane domain and a small cytoplasmic domain. [47]. In its transmembrane form, CX3CL1 is constitutively associated and forms oligomers that are involved in cell adhesion [48]. Membrane-bound CX3CL1 is cleaved by several enzymes, such as the metalloproteinases A disintegrin and metalloproteinase member 10 and 17 (ADAM10 and ADAM17), matrix metalloproteinase 3 (MMP-3), and cathepsin S (reviewed in [49]). The expression of these enzymes can be regulated by hormones or cytokines [50,51]. Changes in the expression of these enzymes in response to viral infection have also been reported. For example, in human small airway epithelial cells, influenza virus infection induces an increase in ADAM17 mRNA levels [52], and Epstein–Barr virus ZTA upregulates the expression of MMP-3 [53]. RSV has also been shown to regulate cathepsin S (among other proteases) in mice [54]. Whether changes in the expression of these proteases caused by RSV infection are responsible for changes in the activity of the CX3CR1-CX3CL1 axis remains to be investigated. 

The soluble CX3CL1 molecule has a chemotactic effect on various cells of the immune system [55]. Enzymes in the cellular microenvironment that enable the release of the chemokine domain finely regulate the balance between CX3CL1-mediated adhesion and chemotactic functions [56,57].

One of the most studied functions of CX3CL1 is its participation in the extravasation of leukocytes through the activated endothelium [45,58]. In contrast to other chemokines, CX3CL1 binds with high affinity to a single receptor, CX3CR1, and this ligand specificity is desirable for drug targets [58,59]. The binding of the chemokine to its seven transmembrane receptor activates a heterotrimeric Gi/o protein and initiates signaling events that trigger diverse cellular responses [55].

However, it has been reported that CX3CL1 can signal and induce its own expression [60]; although this function has been scarcely documented, it could be of great biological relevance and increase the complexity of the system.

## 4. CX3CR1 and RSV Infection 

It is well established that CX3CR1 is an RSV co-receptor. In vitro experiments indicated that the RSV G protein binds to the CX3CR1 receptor on primary human epithelial cells isolated from the airway and promotes infection [32]. Viral G protein binding is specific, as treatment with the anti-CX3CR1 antibody decreases infection. Johnson et al. showed that the binding of the G protein to CX3CR1 is mediated by a CX3C motif in the G protein, since the inclusion of an alanine residue in this motif reverses the effect observed in the wild-type protein [32]. Similar findings were found using primary epithelial cells isolated from pediatric airways and infected with RSV, in which blocking the interaction of RSV with CX3CR1 decreased the viral load [10]. Accumulating evidence from several experimental studies indicates that the conserved CX3C motif facilitates virus binding to CX3CR1^+^ epithelial cells [42], indicating that CX3CR1 is a co-receptor molecule for RSV.

The role of CX3CR1 in RSV infection becomes clinically relevant since CX3CR1 is expressed in the airway of the pediatric population at variable levels [10]. It is important to establish whether the biological variation in CX3CR1 expression is related to the susceptibility or outcome of RSV infection in the pediatric population.

Although CX3CR1 is a co-receptor molecule for RSV entry into epithelial cells, and its expression favors infection, the results from mice which are genetically deficient in CX3CR1 indicate that the absence of this receptor does not protect against the disease; in contrast, the absence of this receptor is related to more severe disease [61]. Newborn mice genetically deficient in CX3CR1 (CX3CR1^−/−^) show higher mortality than wild-type mice, increased mucus production, and increased neutrophil and γδ T cells infiltration in the lungs after RSV infection [61]. Thus, although CX3CR1 favors the entry of the RSV into epithelial cells and its presence implies an increased viral load, the interaction of CX3CR1 with its natural ligand (or with the CX3C domain of the RSV G protein) seems to be relevant for the immune response against the virus. When systemic dissemination occurs, CX3CR1 expression may worsen outcomes. Systemic manifestations of RSV are infrequent but can be very severe [3,4]. Extrapulmonary manifestations of RSV infection include cardiac and neurological alterations, and hepatitis [3,4]. RSV has been detected in the nervous system and heart [62], whereas CX3CR1 is expressed by cardiomyocytes, hepatocytes, and glial cells (proteinatlas.org) [63]. Therefore, it is possible that CX3CR1 facilitates viral transmission in these cells during RSV infection, thereby aggravating the disease. 

## 5. Immune Response against RSV

Once RSV enters the airway epithelial cell (AEC), diverse viral pathogen-associated molecular patterns (PAMPs) can be recognized by multiple host pattern recognition receptors (PRRs) [16,22]. RSV genomic material can be recognized by the retinoic acid inducible gene I (RIG-I) sensor in the cell cytoplasm or by Toll-like receptors (TLRs), such as TLR-3, in endosome compartments, leading to the activation of interferon regulatory (IRF)-3, AP1, and NF-kB transcription factors [64]. Other TLRs are important in RSV infection including TLR-2, -4, and -6 [16,64].

In addition, alveolar macrophages and dendritic cells are stimulated by upon RSV infection [16,64]. The activation of multiple cell types induces the expression of several molecules that control viral infection. Type-I interferons (IFN-I) play a major role in restricting viral replication and dissemination by promoting an antiviral state through the induction of several interferon-stimulated genes in neighboring cells [65,66]. In addition, several proinflammatory mediators are also induced, including cytokines (TNFα, IL-1β, IL-6, INF-λ1, INF-λ2), chemokines (CXCL6, CXCL8 (IL-8), CXCL10, and CX3CL1) [67], and adhesion molecules (ICAM-1 and VCAM-1), which in turn promote immune cell recruitment to control the viral infection. Moreover, major histocompatibility complex molecules (MHC)-I and II are also induced, favoring the onset of an adaptive immune response against the pathogen [11,68]. 

The recruitment of neutrophils is an early event following RSV infection; these cells are the most abundant leukocytes that infiltrate the lungs upon infection [69]. Neutrophil activation induces neutrophil extracellular traps (NETs) which may trap viral particles that, in turn, can be inactivated by diverse neutrophil secreted molecular mediators such as defensins; in fact, antimicrobial peptides such as cathelicidin and defensins produced by AECs may also restrict RSV replication [70,71]. Viral particles can be phagocytosed by neutrophils and destroyed by distinct mechanisms, including respiratory burst (ROS production) [69]. Nevertheless, excessive neutrophil responses may contribute to lung injury.

NK cells respond to viral infection; thus, they are also important in the control of RSV infection. These cells kill the infected cells, limiting viral replication, and produce IFNγ, promoting a Th1 type response [11]. A reduction in perforin secretion in RSV-infected NK cells has been demonstrated, which might be an evasion mechanism of RSV against the host immune response [72]. Furthermore, RSV induces an upregulation of inhibitory receptors in NK cells.

Dendritic cells (DCs), considered the bridge between innate and adaptive immunity, are also recruited upon RSV infection. Plasmacitoid DCs (pDCs) are a subset of DCs specialized in high production of IFN-I, a major contributor to the antiviral response. pDCs recognize viral RNA through TLR-7 and -8, leading to downstream activation of IRF-7, AP1, and NFκB, which in turn initiate an increased production of IFN-I [65,66]. 

Moreover, pDCs are required for adequate RSV-specific cytotoxic T cell generation [68]. Nevertheless, RSV infection counteracts IFN-I production by pDCs [73]. In addition, conventional DCs (cDCs) are also recruited to the site of infection, where they acquire viral antigens by direct infection or phagocytosis of infected cells. RSV sensing by TLRs and RLRs (RIG-1 like receptors, which detect RNA from viruses in the cytoplasm) in cDCs induces their maturation with the upregulation of MCH and co-stimulatory molecules, and the secretion of several Th-1 type cytokines. cDCs initiate an adaptive immune response against RSV upon antigen presentation to specific T cells. RSV infection, however, induces cDC dysfunction that alters their capacity to stimulate T cells [74].

Similar to other viral infections, adaptive immunity against RSV is essential for disease control. Both CD4^+^- and CD8^+^-specific T cell responses, as well as high titers of virus-specific antibodies, are required for disease control [64]. Severe and/or persistent RSV infections in T cell-immunodeficient patients, such as those living with HIV/AIDS, account for the fundamental role of T cell responses in RSV elimination [16,64]. The activation and differentiation of CD4^+^ T cells into the Th1 phenotype correlates with protective responses and efficient viral clearance. In addition, CD4^+^ T cells are required for adequate B cell activation and the generation of plasmablasts and memory B cells [22]. Cytotoxic CD8^+^ T cells are fundamental for RSV clearance, supported by findings from post-mortems of fatal cases of RSV in which few CD8^+^ T cells were found in the lung tissue [75].

Upon RSV infection, B cell activation, proliferation, and differentiation into plasmablasts leads to antibody production. Neutralizing antibodies against the viral F protein are the major contributors to infection control, and memory B cells are also generated [11,16]. In newborns, natural IgM antibodies play a central role against RSV infection, and the transfer of maternal RSV-specific IgG antibodies is another way to achieve protection against infection [76]. From three years old onwards, the generation of diverse neutralizing antibodies, mainly against the F viral protein [16,64], and switching to IgG and IgA isotypes correlate with reduced morbidity and protection [12]. Regulatory T cells (Tregs) play a central role in controlling disease severity during RSV infection by limiting immunopathology [11,77].

RSV has evolved several mechanisms to evade the host’s immune response. The non-structural viral proteins NS1 and NS2 are the first to be transcribed upon infection and are both potent IFN-I inhibitors. Deletion mutants of these proteins generate attenuated viruses, which have been proposed as vaccines [66,78]. Both proteins affect the RIG-I signaling pathway to inhibit IFN-I production; NS1 binds the mitochondrial antiviral-signaling protein, interfering with its interaction with RIG-I, while NS2 interacts with the N-terminal caspase activation and recruitment domain of RIG-I, thus inhibiting its downstream activation pathway. Indeed, NS1 is considered a multifunctional RSV protein that counteracts the immune response in distinct ways [79,80,81,82]. Evasion mechanisms elicited by RSV include the modulation of innate immune responses, such as the activation of DCs, macrophages, and NK cells, the induction of Tregs, and antigenic variation. RSV exhibits high antigenic and genetic diversity, particularly in the G protein, which is a target of the host immune response. This allows RSV to evade immune recognition and establish reinfections [28]. Alterations in the CX3CL1-CX3CR1 axis are also involved in the immune evasion mechanisms elicited by RSV. The RSV G protein is involved in IFN-I antagonism, as detailed in the following section [66].

Recent research has highlighted the role of the CX3CL1-CX3CR1 axis in the host immune response to RSV infection.

## 6. CX3CR1-CX3CL1 Contribution to the Immune Response against RSV 

As mentioned above, infection of AECs with RSV increases the expression of several cytokines including CX3CL1. When cells are infected with a mutant RSV that has an insertion in the CX3C motif of the G protein or when cells are treated with an anti-CX3CR1 antibody prior to infection with wild-type RSV, the induction of cytokines is decreased [67]. These results indicate that binding of the CX3C motif of the RSV G protein to CX3CR1 influences cell activation and host immune responses. 

Most immune cells relevant in the resolution of RSV infection express CX3CR1 receptors, such as monocytes, macrophages, DCs, NK cells, and CD8^+^ T cells [83]. During RSV infection, these cells are recruited to the site of infection and play an important role in virus elimination [84]. For example, CX3CR1 is highly expressed in non-classical monocytes, which are normally associated with viral sensing and clearance [85,86]. In the context of RSV pathophysiology, monocytes are recruited through chemokines secreted by infected epithelial cells in the replicative phase of infection, and they are important for limiting the infection [87,88]. People carrying the M280 polymorphic variant of CX3CR1 tend to present with more severe cases of bronchiolitis in response to RSV [89]. The CX3CR1 M280 variant has been associated with a decreased adhesive capacity of human monocytes [90]; hence, patients homozygous for CX3CR1 M280 may experience a decreased transmigration of monocytes and other immune system cells to the infected lung, which may contribute to the reported severe cases of bronchiolitis. In addition, this polymorphic variant may be associated with a decreased ability to prevent death by apoptosis in response to receptor stimulation with CX3CL1 [91]. A decrease in adhesion and survival could imply that the antiviral activity associated with monocytes is diminished.

Notably, alterations in the CX3CL1-CX3CR1 axis are also involved in the immune evasion mechanisms elicited by RSV. The soluble form of the RSV G-protein, when binding to CX3CR1, acts as a competitive antagonist ligand, interfering with the chemotactic responses of immune cells and preventing their recruitment to the site of infection [41,92]. Consequently, there is a decrease in both the recruitment of CX3CR1-expressing T cells and NK cells [93], and the concentration of cytokines such as IFN-γ, IL-4, and CX3CL1 in the pulmonary microenvironment. Deficient IFN-γ production hinders the host antiviral response [93]. Moreover, because of the RSV G CX3C motif and CX3CR1 interaction, the production of type I and III IFNs by epithelial cells and IFN-α production by pDCs is reduced compared to a G protein with a mutated CX3C motif or when a monoclonal antibody against RSV G is used to prevent interaction with CX3CR1 [66,94]. Similar behavior has been observed in CD4^+^ T central memory and CD8^+^ T effector memory cells, which produce less IFN-γ when stimulated with RSV G [94]. Hence, the displacement of CX3CL1 by the RSV G protein affects both innate and adaptive immune cells, antagonizing IFN production in favor of RSV replication. Furthermore, during RSV infection, non-classical monocytes are recruited to the site of infection where they play an important role in virus elimination. Nevertheless, these cells can be productively infected with RSV, which limits their antiviral function [11,95].

Regulatory B cells inhibit the immune response through the secretion of cytokines, such as IL-10, IL-35, and tumor growth factor-β (TGF-β), and by direct interactions with other cells through molecules such as CD19, CD80, and CD86 [96,97]. These signals decrease the activation of effector cells and induce the differentiation of CD4^+^ T cells to regulatory T cells, which indirectly increases the severity of infection. Recently, Zhivaki et al. found a new subtype of regulatory B cells restricted to infants aged a few months (neonatal regulatory B cells, nBReg) [98]. In these cells, IgM recognizes RSV, inducing the expression of CX3CR1 through BCR signaling. Subsequently, RSV infects nBreg through CX3CR1 and replicates with high efficiency while IL-10 is produced [98]. Thus, infection of nBregs results in a direct increase in viral load and a decrease in effector cell activity. These findings are of great importance in explaining the clinical outcomes of pediatric patients hospitalized for RSV. The authors found a higher frequency of nBReg in patients who required supplemental oxygenation and spent more days in intensive care, and concluded that nBReg cells are a determining factor for the development of a severe RSV infection [96,98]. Figure 1 summarizes the consequences of the interaction of CX3CL1 and the G protein with CX3CR1 during RSV infection. 

Different types of vaccines have been developed against RSV infection, and as of August 2024, three have been approved for use in the general population. All of these are directed against the F protein [99]. In addition, therapeutic alternatives with diverse viral targets continue to be researched owing to the complexity of RSV infection. The G protein is known to induce protective antibodies against RSV. In a recent report, a new vaccine was tested using the CX3C-motif mutated regions of the RSV G protein as the immunogen [100]. The mutated proteins were more immunogenic for antibody production in BALB/c mice immunized with these molecules than proteins with the wild type CX3C-motif. Anti-G protein antibodies were effective in blocking the interaction of the CX3CR1 receptor with the RSV G protein without blocking the interaction between CX3CR1 and CX3CL1 [100]. This experimental approach is a promising starting point for the development of an effective vaccine that generates antibodies capable of reducing RSV infection through the blocking of the CX3CR1 co-receptor, but at the same time without affecting the natural functions of CX3CL1 and CX3CR1. 

## 7. Future Directions

Recent studies suggest that the RSV G protein, despite exhibiting high variability compared to the F protein [90], could be a prophylactic and therapeutic target for RSV infection in humans. In a murine model of RSV infection, Lee et al. demonstrated that treatment with monoclonal antibodies directed against different regions of the RSV G protein reduced lung pathology and infectious viral titers [101]. Despite the high variability of the G protein among the different strains, the antibodies tested were able to confer broad-spectrum protection; that is, they worked against both the RSV-A2 and RSV-B1 strains. Additionally, passive immunization with anti-G antibodies enhances protection when used in conjunction with anti-F antibodies [101]. The antibodies characterized in this study are very promising, and it is important to test them in different experimental models, to eventually move on to their pre-clinical evaluation.

Moreover, another recent study characterized the stability of the binding between the conserved cysteine-noose G domain (cdnG) of the G protein (which includes the CX3C motif), and a canonical CX3CL1-binding site at the amino-terminal end found in natural variants of CX3CR1 [102]. The results showed that the cdnG complexes and different polymorphic variants of CX3CR1 have similar binding stabilities. In addition, important amino acid residues were identified, such as Glu 279, which is found in the second extracellular loop of CX3CR1 and is important for stabilizing the binding of the complex, and Glu 174, which promotes hydrogen bonding [102]. These data are important for the design of peptides that interfere with the binding between cdnG and CX3CR1, and suggest that such peptides could have a similar effect in individuals with different polymorphic variants of CX3CR1.

## 8. Conclusions

The RSV G protein binds to the CX3CR1 receptor, which then functions as a viral co-receptor in several cell types. CX3CR1 has clinical relevance since it is expressed in the airways of the pediatric population, and its expression has been related to a higher number of infected epithelial cells. The interaction between CX3CL1 and CX3CR1 plays a crucial role in the immune response to RSV infection because mice genetically deficient in CX3CR1 develop more severe cases of infection. The presence of the CX3C motif in the RSV G protein during RSV infection decreases the recruitment of cells expressing CX3CR1, such as T cells and NK cells, which are involved in virus clearance. Moreover, the soluble form of the RSV G protein competes with CX3CL1 to bind to CX3CR1. 

In contrast, the binding of CX3CR1 to the CX3C motif of RSV can modulate the antiviral immune response by inducing the anti-inflammatory cytokine IL-10, which blocks the antiviral response. If confirmed, this is a novel mechanism for the evasion of the host’s immune response. 

Finally, mutations in the CX3C motif of the G protein increase its antigenic immunogenicity and have been used to induce antibodies in animal models that do not prevent the binding of CX3CR1 to CX3CL1. 

Further research into the roles of CX3CL1 and CX3CR1 in RSV pathogenesis may lead to the development of novel treatment therapies and prevention strategies targeting the host’s immune response to RSV infection, as well as to the development of new vaccines.

## Figures and Tables

**Figure 1 ijms-25-09800-f001:**
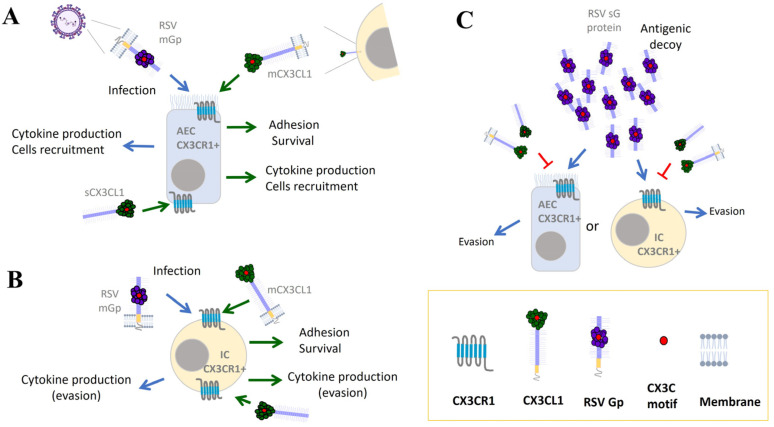
The presence of RSV G protein alters the natural functions of the host’s immune response. The G protein in RSV binds to the CX3CR1 receptor expressed on the membrane of target cells, such as airway epithelial cells (AEC), and enables infection. This binding occurs through the CX3C domain of the G protein, which mimics the CX3C motif of the chemokine CX3CL1. The complexity of the system is increased because cellular responses to membrane-bound or soluble CX3CL1 binding are also different, and RSV G protein exists in bound and secreted forms (**A**). Various CX3CR1^+^ immune cells, such as monocytes, dendritic cells, NK cells, and some subtypes of T and B cells, contribute to the immune response against RSV. These cells can be infected with RSV and contribute to viral evasion mechanisms (**B**). RSV G protein functions as an antigenic decoy and shifts the binding of CX3CL1 to CX3CR1 (**C**).

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
