# Peer review of "The Role of the CX3CR1-CX3CL1 Axis in Respiratory Syncytial Virus Infection and the Triggered Immune Response"

_ijms, 2024, doi:10.3390/ijms25189800_

Round 1

Reviewer 1 Report

Comments and Suggestions for Authors

The article reviews the role of the CX3CR1-CX3CL1 axis in RSV infection and the immune response it triggers. It mentions the mechanisms by which RSV exploits the CX3CR1 receptor to facilitate infection. The article also highlights recent advancements in understanding RSV pathogenesis, particularly the significance of the CX3C motif in the viral G protein and its interaction with CX3CR1. The article has several flaws that must be corrected in the present form.

Mayor concerns

-       The abstract must be improved by adding more relevant data. 

-       Lines 13 and 24, Respiratory syncytial virus.

-       Lines 24-34 must be improved by adding data on the clinical manifestations and the effect of niversimab on the cases of hospitalizations.

-       Lines 35-38 must be improved by adding more references, enriching the research context, and including the genotypes of RSV, a crucial aspect for a comprehensive understanding of the virus.

-       After a relevant statement, the authors must cite the proper reference.

-       Section 2 must be improved. The author must add more recent and relevant data about the infection cycle and detail the role of each protein. Nucleolin is not mentioned.

-       Line 80: Define ADAM10, ADAM17, and MMP-3. Besides, several abbreviations have yet to be described in the text at the first mention.

-       Lines 81-84 have no references.

-       Lines 90-94 have no references.

-       Detail the meaning of CX3CR1 M280 for the immune response.

-       Detail the relevance of Breg for the RSV infection.

-       The authors must update the data regarding the vaccines and possible treatments against RSV.

-       Overall, the review needs more discussion and proposal of new ideas to address the role of CX3CR1-CX3CL1 or discussion about current evidence. There is no critical view of the literature.

-       More detail is needed in each section.

-       The Figure is too simple and lacks visual appeal. The authors must draw the cells, RSV, and molecules correctly.

Minor concerns

-       The grammar must be reviewed extensively and corrected.

-       Line 222: IFN-g.

-       Correct the wrong use of capitalization.

-       More actual references are needed.

Comments on the Quality of English Language

The English quality is fine, but grammar is not.

Reviewer 2 Report

Comments and Suggestions for Authors

This review from Rivas-Fuentes et al summarizes current literature concerning the multi faceted contribution of CXC3R1-CXCL1 towards RSV infection.  CXC3R1 has been found to be a co-receptor for RSV cell entry, and limiting its expression or bioavailability in cell culture significantly reduces RSV infection.  However, in-tact CXC3R1-CXCL1 signaling is important for controlling RSV infection in-vivo, which is believed to be due to a requirement for CXC3-mediated immune cell recruitment.  Outside of many immune cells expressing CXC3R1, this claim is strengthened by CXC3R1-RSV G protein binding increasing cytokine expression, secreted RSV G acting as a competitive inhibitor to CXC3R1 on immune cells, and human patients deficient in CXC3R1 signaling having more severe RSV pathology.

As a whole, the review is well-written and highlights an interesting nuance to RSV infection that is important to consider when developing new therapeutics against RSV.  I only have a few minor comments for the authors to address prior to acceptance:

1) In lines 74-85, the authors should clarify events that lead to expression of enzymes that cleave membrane-bound CXC3L1 and how those events are affected by viral infection.

2) In lines 115-126, the authors indicate that CXC3R1 deficiency in mice is associated with increased morbidity and mortality to RSV infection as well as higher immune cell infiltrate at the time of morality, but they should include details on how CXC3R1 may affect systemic infection since CXC3R1 likely has different expression profiles in different tissue types.

3) The CXC3 axis has also been found to be critical for CD8 T cell entry into vaccinia-infected skin (Hickman et al, Immunity 2015)- the authors should indicate whether any similar study has also been done on CD8 T cells or adaptive immune cells as a whole in lungs.

Author Response

July 26, 2024

Corrections for IJMS

Title: The Role of the CX3CR1-CX3CL1 Axis in Respiratory Syncytial Virus Infection and the Triggered Immune Response

Author(s): Selma Rivas-Fuentes*, Teresa Santos-Mendoza, Alfonso Salgado-Aguayo and Edgar Sevilla-Reyes

ID: ijms-3117624

 Dear Reviewers,

The authors appreciate the time you dedicated to reviewing our work, and we are thankful for your valuable comments, which have enhanced it.

We have made the corresponding changes in the text, and we have addressed your observations below.

Comments for Reviewer 2:

            “As a whole, the review is well-written and highlights an interesting nuance to RSV infection that is important to consider when developing new therapeutics against RSV.  I only have a few minor comments for the authors to address prior to acceptance…”

  1. In lines 74-85, the authors should clarify events that lead to expression of enzymes that cleave membrane-bound CXC3L1 and how those events are affected by viral infection.

Dear reviewer, thank you for your commentary.

The expression of the proteases that are able to shed CX3CL1 can be regulated by hormones or cytokines; for example ADAM 10 expression is upregulated by dihydrotestosterone, IGFI and  EGF (McCulloch, 2004), while their activity can also be regulated by external factors, such as thrombin, which has been shown to stimulate the activity of ADAM17 (Arora,2008). More importantly, the regulation of the expression of these enzymes in response to viral infections has been detected: in human small airway epithelial cells, influenza virus infection induces an increase in ADAM17 mRNA (Schweitzer, 2021), and both Hepatitis B virus X protein (Yu, 2005) and Epstein-Barr virus ZTA (Lan, 2013) upregulated the expression of MMP-3. RSV has also been shown to regulate cathepsin S (among other proteases) in mice (Foronjy, 2014), and the bovine RSV plus H. somni synergistically upregulated the expression of MMP3 (Agnes, 2013). It remains to be investigated whether changes in the expression of these proteases caused by RSV infection are responsible for changes in the activity of the CX3CR1-CX3CL1 axis.

To address your concern, we have included the following paragraph (lines 74 to 81):

… The expression of these enzymes can be regulated by hormones or cytokines (McCulloch, 2004, Arora, 2008). Changes in the expression of these enzymes in response to viral infections have been detected. For example, in human small airway epithelial cells, influenza virus infection induces an increase in ADAM17 mRNA (Schweitzer, 2021), and Epstein-Barr virus ZTA (Lan, 2013) upregulated the expression of MMP-3. RSV has also been shown to regulate cathepsin S (among other proteases) in mice (Foronjy, 2014). It remains to be investigated whether changes in the expression of these proteases caused by RSV infection are responsible for changes in the activity of the CX3CR1-CX3CL1 axis.

References:

 McCulloch, D. R., Akl, P., Samaratunga, H., Herington, A. C., & Odorico, D. M. (2004). Expression of the disintegrin metalloprotease, ADAM-10, in prostate cancer and its regulation by dihydrotestosterone, insulin-like growth factor I, and epidermal growth factor in the prostate cancer cell model LNCaP. Clinical cancer research, 10(1), 314-323.

Arora, P., Cuevas, B. D., Russo, A., Johnson, G. L., & Trejo, J. (2008). Persistent transactivation of EGFR and ErbB2/HER2 by protease-activated receptor-1 promotes breast carcinoma cell invasion. Oncogene, 27(32), 4434-4445

-Schweitzer, K. S., Crue, T., Nall, J. M., Foster, D., Sajuthi, S., Correll, K. A., ... & Petrache, I. (2021). Influenza virus infection increases ACE2 expression and shedding in human small airway epithelial cells. European Respiratory Journal, 58(1).

-Yu, F. L., Liu, H. J., Lee, J. W., Liao, M. H., & Shih, W. L. (2005). Hepatitis B virus X protein promotes cell migration by inducing matrix metalloproteinase-3. Journal of hepatology, 42(4), 520-527.

Lan, Y. Y., Yeh, T. H., Lin, W. H., Wu, S. Y., Lai, H. C., Chang, F. H., ... & Chang, Y. (2013). Epstein-Barr virus Zta upregulates matrix metalloproteinases 3 and 9 that synergistically promote cell invasion in vitro. PloS one, 8(2), e56121.

Foronjy, R. F., Dabo, A. J., Taggart, C. C., Weldon, S., & Geraghty, P. (2014). Respiratory syncytial virus infections enhance cigarette smoke induced COPD in mice. PloS one, 9(2), e90567.

Agnes, J. T., Zekarias, B., Shao, M., Anderson, M. L., Gershwin, L. J., & Corbeil, L. B. (2013). Bovine respiratory syncytial virus and Histophilus somni interaction at the alveolar barrier. Infection and immunity, 81(7), 2592-2597.

  1. In lines 115-126, the authors indicate that CXC3R1 deficiency in mice is associated with increased morbidity and mortality to RSV infection as well as higher immune cell infiltrate at the time of morality, but they should include details on how CXC3R1 may affect systemic infection since CXC3R1 likely has different expression profiles in different tissue types

Dear reviewer:  We thank you for your comment.

Although CX3CR1 facilitates RSV entry into airway epithelial cells, its deficiency does not inhibit infection, and has an adverse effect on the outcome, meaning that the CX3CR1-CX3CL1 axis is important for resolution of infection. However, when systemic dissemination of RSV occurs, the outcome might worsen. Systemic manifestations of RSV are infrequent, but become very severe (Eisenhut, M, 2006, Bottino, 2021), and in these cases CX3CR1 could aggravate the infection. For example, among the extrapulmonary manifestations reported for RSV are various cardiac complications such as cardiac failure with hypotension, myocardial damage and arrhythmias; neurological involvement and hepatitis (Eisenhut, M, 2006). RSV has even been detected on nervous system and heart (Gkentzi, 2018)

CX3CR1 is expressed in many types of epithelial cells and endothelium, thus CX3CR1 can potentially be expressed in all tissues of the human body. Additionally, it is expressed by different cells of the immune system that can infiltrate tissues (Lee, 2018). Interestingly, according to the information available in the Human Protein Atlas portal (www.proteinatlas.org), protein level expression of CX3CR1 has been detected in cardiac muscle (cardiomyocytes), liver (hepatocytes), brain (glia cells), among others. Therefore, it is possible that during RSV infection, CX3CR1 facilitates the infection of cardiomyocytes, hepatocytes, and glia cells, in some of these organs (Gkentzi, 2018) contributing to the systemic manifestations of RSV infection.

In order to address your concern, we have included the following paragraph and references (line 119-126):

When systemic dissemination has occurred, CX3CR1 expression could worsen the outcome. Systemic manifestations of RSV are infrequent, but can be very severe (Eisenhut, M, 2006, Bottino, 2021). Extrapulmonary manifestations of RSV infection include cardiac and neurological alterations, and hepatitis (Eisenhut, M, 2006, Bottino, 2021). RSV has been detected on the nervous system and heart (Gkentzi, 2018), while CX3CR1 is expressed by cardiomyocytes, hepatocytes and glia cells (www.proteinatlas.org). Therefore, it is possible that during RSV infection, CX3CR1 facilitates viral transmission in these cells, aggravating the disease.

References:

Eisenhut, M. (2006). Extrapulmonary manifestations of severe respiratory syncytial virus infection–a systematic review. Critical Care, 10, 1-6.

Bottino, P., Miglino, R., Pastrone, L., Barbui, A. M., Botta, G., Zanotto, E., ... & Cavallo, R. (2021). Clinical features of respiratory syncytial virus bronchiolitis in an infant: rapid and fatal brain involvement. BMC pediatrics, 21, 1-5.

Gkentzi, D., Dimitriou, G., & Karatza, A. (2018). Non-pulmonary manifestations of respiratory syncytial virus infection. Journal of Thoracic Disease, 10(Suppl 33), S3815.

Lee, M., Lee, Y., Song, J., Lee, J., & Chang, S. Y. (2018). Tissue-specific role of CX3CR1 expressing immune cells and their relationships with human disease. Immune network, 18(1).

Sjöstedt, E., Zhong, W., Fagerberg, L., Karlsson, M., Mitsios, N., Adori, C., ... & Mulder, J. (2020). An atlas of the protein-coding genes in the human, pig, and mouse brain. Science, 367(6482), eaay5947.

  1. The CXC3 axis has also been found to be critical for CD8 T cell entry into vaccinia-infected skin (Hickman et al, Immunity 2015)- the authors should indicate whether any similar study has also been done on CD8 T cells or adaptive immune cells as a whole in lungs.

Dear reviewer, thank you very much for your comment. In the vaccinia virus infection, it has been reported that CD8+ T cells, through the chemokine receptor CXCR3 (the receptor for CXCL9 and CXCL10 chemokines), target the infected skin cells and manage virus clearance (Hickman et al, 2015).

To attend your commentary, we have included the followed information (line 217-220):

For example, CX3CR1 is highly expressed on non-classical monocytes, which are normally associated to viral sensing and clearance [58,59]. CX3CR1 could then play a similar role to the one CXC3R (the receptor for CXCL9 and CXCL10) has during vaccinia infection, allowing CD8+ T cells to target the infected skin cells [60] 

Hickman, H. D., Reynoso, G. V., Ngudiankama, B. F., Cush, S. S., Gibbs, J., Bennink, J. R., & Yewdell, J. W. (2015). CXCR3 chemokine receptor enables local CD8+ T cell migration for the destruction of virus-infected cells. Immunity, 42(3), 524-537.

Reviewer 3 Report

Comments and Suggestions for Authors

Abstract:

In the abstract please delete "and pharmacological treatments". There are no pharmacological treatments for RSV disease which have been shown to be effective in randomized controlled trial meta-analyses.

Introduction:

The paragraph: RSV, formally named as Orthopneumovirus hominis [3], is a negative-sense, single- 35 stranded RNA virus (Baltimore Group V) belonging to the family Pneumoviridae, genus 36 Orthopneumovirus. Two different subtypes of RSV have been described, RSV-A and RSV- 37 B [4]. 38 RSV is an enveloped virus, with a lipid bilayer derived from the host cell apical mem- 39 brane. The virion is pleomorphic, ranging from nearly spherical to filamentous, a shape 40 that is mostly found in cell cultures, with a diameter of approximately 150-300 nm. The 41 genome of RSV is non-segmented and approximately 15.2 kb in length, with 10 ORFs, 42 encoding 11 proteins: NS1, NS2, N, P, M, SH, G, F, M2.1, M2.2 and L" is redundant and should be deleted.

The following paragraph should also be deleted:"Besides N and P (...) negative sense RNA genomes".

In the paragraph on biological aspects the sentence: "It is characterised (...)cysteins" should be deleted as irrelevant.

Line 81: Which cells of the immune system? If there is an analysis of neutrophil leucocyte involvement please fully expand.

Delete: "In a pathophysiological context 86 the adhesive function of CX3CL1 is relevant in atherosclerotic plaque formation, through monocyte binding [27]. In the central nervous system, CX3CL1 is robustly expressed by neurons, and its elicited signaling are relevant in cerebral ischemia, epilepsy and in several neurodegenerative diseases [28], although its role in these diseases remains controversial."

Please expand fully:"Alterations in the CX3CL1-CX3CR1 axis are also involved in the immune evasion mechanisms"

Line 230The sentence: In addition "human monocytes with the CX3CR1 M280 variant are known to be deficient in CX3CL1- 231 mediated survival" is unclear: Please rephrase and explain.

Comments on the Quality of English Language

The authors need to have their manuscript proof read by a native English speaker.

Author Response

   July 26, 2024

Corrections for IJMS

Title: The Role of the CX3CR1-CX3CL1 Axis in Respiratory Syncytial Virus Infection and the Triggered Immune Response

Author(s): Selma Rivas-Fuentes*, Teresa Santos-Mendoza, Alfonso Salgado-Aguayo and Edgar Sevilla-Reyes

ID: ijms-3117624

 Dear Reviewers,

The authors appreciate the time you dedicated to reviewing our work, and we are thankful for your valuable comments, which have enhanced it.

We have made the corresponding changes in the text, and we have addressed your observations below.

       Comments for Reviewer 3:

  1. In the abstract, please delete "and pharmacological treatments". There are no pharmacological treatments for RSV disease which have been shown to be effective in randomized controlled trial meta-analyses.

 Dear reviewer, thank you for your commentary. We have deleted this information. 

  1. The paragraph: RSV, formally named as Orthopneumovirus hominis [3], is a negative-sense, single- 35 stranded RNA virus (Baltimore Group V) belonging to the family Pneumoviridae, genus 36 Orthopneumovirus. Two different subtypes of RSV have been described, RSV-A and RSV- 37 B [4]. 38 RSV is an enveloped virus, with a lipid bilayer derived from the host cell apical mem- 39 brane. The virion is pleomorphic, ranging from nearly spherical to filamentous, a shape 40 that is mostly found in cell cultures, with a diameter of approximately 150-300 nm. The 41 genome of RSV is non-segmented and approximately 15.2 kb in length, with 10 ORFs, 42 encoding 11 proteins: NS1, NS2, N, P, M, SH, G, F, M2.1, M2.2 and L" is redundant and should be deleted. The following paragraph should also be deleted:"Besides N and P (...) negative sense RNA genomes".

Dear reviewer:  We thank you for your comment. To attend to the reviewer’s comment, the paragraph was simplified and focused on key information required for the reader to visualize the context. Bellow we showed the paragraph with corrections. In addition, we eliminated the information that is indicated in the point 2.2.

Paragraph with corrections:

  • RSV is a pleomorphic enveloped virus with a virion diameter ranging from 150 to 300 nanometers. Its genome is a non-segmented, negative-sense, single-stranded RNA of 15.2 kilobases, comprising 10 ORFs, encoding 11 proteins [5]. RSV-A and RSV-B are two subtypes that reflect G protein diversity [4].

  • We deleted reference 3: Walker, P.J.; Siddell, S.G.; Lefkowitz, E.J.; Mushegian, A.R.; Adriaenssens, E.M.; Alfenas-Zerbini, P.; Dempsey, D.M.; Dutilh, B.E.; García, M.L.; Curtis Hendrickson, R.; et al. Recent Changes to Virus Taxonomy Ratified by the International Committee on Taxonomy of Viruses (2022). Virol. 2022, 167, 2429–2440, doi:10.1007/s00705-022-05516-5. We deleted the following paragraph "Besides N and P (...) negative sense RNA genomes".

  1. In the paragraph on biological aspects the sentence: "It is characterised (...)cysteins" should be deleted as irrelevant

        Dear reviewer, thank you very much for your comment. Unfortunately, we omitted in our write-up  the importance of the CX3C domain from CX3CL1. To address your comment, at the end of the section that precedes the sentence “It is characterized by being the only chemokine to have three amino acid residues between the two conserved cysteines” (line 62-63), we made the following changes:

The statement: “Glycoprotein G binds to the surface of the host cell, binding to CX3CR1 mimicking the host chemokine CX3CL1 (line 58-59).” was changed to:

“Among the proteins that constitute RSV, of particular importance is Glycoprotein G. This protein interacts with the surface of the host cell through binding to the CX3C motif of the chemokine CX3CL1, which, as will be explained in later sections, is important for both infection and host immune response (line 58-61).”

  1. Line 81: Which cells of the immune system? If there is an analysis of neutrophil leucocyte involvement, please fully expand.

Dear reviewer, we have detailed the cells of the immune system in section 6 (line 215)

Most of the immune cells relevant in the resolution of RSV infection express the CX3CR1 receptor, such as monocytes, macrophages, DCs, NK, and CD8+ T lymphocytes (Lee et. al. 2018)

 Lee, M.; Lee, Y.; Song, J.; Lee, J.; Chang, S.-Y. Tissue-Specific Role of CX3CR1 Expressing Immune Cells and Their Relationships with Human Disease. Immune Netw. 2018, 18, e5, doi:10.4110/in.2018.18.e5.

  1. Delete: "In a pathophysiological context 86 the adhesive function of CX3CL1 is relevant in atherosclerotic plaque formation, through monocyte binding [27]. In the central nervous system, CX3CL1 is robustly expressed by neurons, and its elicited signaling are relevant in cerebral ischemia, epilepsy and in several neurodegenerative diseases [28], although its role in these diseases remains controversial."

Dear reviewer, thank you for your commentary. We have deleted this information. 

  1. Please expand fully: "Alterations in the CX3CL1-CX3CR1 axis are also involved in the immune evasion mechanisms"

Dear reviewer, thank you for your commentary. To address your recommendation, at the section “CX3CR1-CX3CL1 contribution to the immune response against RSV”, we have included the following information

Notably, alterations in the CX3CL1-CX3CR1 axis are also involved in the immune evasion mechanisms elicited by RSV. Specifically, the CX3C motif in the RSV G protein competes for CX3CR1 with its natural ligand CX3CL1. As a result of RSV G interaction with CX3CR1, the production of type–I and –III IFNs by epithelial cells, and IFN α by pDCs, is reduced compared to a G protein with a mutated CX3C motif or when a monoclonal antibody (mAb) against RSV G is used to prevent interaction with CX3CR1 (Chirkova, 2013; Bergeron, 2023). A similar behavior has been observed in CD4+ T central memory and CD8+ T effector memory cells, which produce less IFN-g under RSV G interaction conditions Chirkova, 2013). Hence, displacement of CX3CL1 interaction with CX3CR1 by RSV G protein affects both innate and adaptive immune cells antagonizing IFN production in favor of RSV replication.

Chirkova, T., Boyoglu-Barnum, S., Gaston, K. A., Malik, F. M., Trau, S. P., Oomens, A. G., & Anderson, L. J. (2013). Respiratory syncytial virus G protein CX3C motif impairs human airway epithelial and immune cell responses. Journal of virology, 87(24), 13466-13479.

To improve readability, we changed the location of one paragraph in that section. For clarity, we have marked the previous paragraph with italics below to show its final position. The new information is shown with bold lettering.

…Most of the immune cells relevant in the resolution of RSV infection express the CX3CR1 receptor, such as monocytes, macrophages, DCs, NK, and CD8+ T lymphocytes [57]. During RSV infection, these cells are recruited to the site of infection, playing an important role in virus elimination [59]; for example, highly-expressing-CX3CR1 non-classical monocytes, normally associated to viral sensing and clearance, are recruited to the site of infection [58]. These cells are the main mediators of the cytotoxic response against RSV, and it has been observed that severe cases of RSV-derived bronchitis present a decrease in CD8+ T lymphocytes [60].

On the other hand, people carrying the polymorphic variant of CX3CR1 M280 tend to present more severe cases of bronchiolitis in response to RSV [62]. The CX3CR1 M280 variant has been associated with decreased adhesive capacity of human monocytes [63], hence patients homozygous for CX3CR1 M280 may have decreased transmigration of monocytes and other immune system cells to the infected lung and this may contribute to the severe cases of bronchiolitis reported. In the context of RSV pathophysiology, monocytes are recruited through chemokines secreted by infected epithelial cells in the replicative phase of infection and they are important to limit the infection [64,65]. In addition, human monocytes with the CX3CR1 M280 variant are known to be deficient in CX3CL1-mediated survival [66], so individuals homozygous for the CX3CR1 M280 variant have a lower number of antiviral monocytes in the lung.

Notably, alterations in the CX3CL1-CX3CR1 axis are also involved in the immune evasion mechanisms elicited by RSV. The secreted form of RSV G-protein, by binding to CX3CR1, acts as a competitive antagonist ligand, interfering with the chemotactic responses of immune cells, and preventing the recruitment of immune cells to the site [35,36]. As a consequence, there is a decrease in both the recruitment of CX3CR1-expressing T lymphocytes and NK cells [61], and the concentration of cytokines such as IFN-g, IL-4 and CX3CL1 in the pulmonary microenvironment. A deficient production of INF-g hinders the host antiviral response [61]. Moreover, because of RSV G CX3C motif and CX3CR1 interaction the production of type–I and –III IFNs by epithelial cells, and IFN-α produced by pDCs is reduced compared to a G protein with a mutated CX3C motif or when a monoclonal antibody against RSV G is used to prevent interaction with CX3CR1 (Chirkova, 2013; Bergeron, 2023). A similar behavior has been observed in CD4+ T central memory and CD8+ T effector memory cells, which produce less IFN-g when they are stimulated with RSV G (Chirkova, 2013). Hence, displacement of CX3CL1 by the RSV G protein affects both innate and adaptive immune cells antagonizing IFN production in favor of RSV replication.  For another hand, during RSV infection, non-classical monocytes are recruited to the site of infection in which they play an important role in virus elimination; nevertheless, these cells can be productively infected by RSV limiting their antiviral function (Agac, 2023 and Miulla)

Chirkova, T., Boyoglu-Barnum, S., Gaston, K. A., Malik, F. M., Trau, S. P., Oomens, A. G., &

              Anderson, L. J. (2013). Respiratory syncytial virus G protein CX3C motif impairs human airway

epithelial and immune cell responses. Journal of virology, 87(24), 13466-13479.

Midulla, F., Huang, Y. T., Gilbert, I. A., Cirino, N. M., McFadden Jr, E. R., & Panuska, J. R. (1989). Respiratory syncytial virus infection of human cord and adult blood monocytes and alveolar macrophages. Am Rev Respir Dis, 140(3), 771-777.

Agac, A., Kolbe, S. M., Ludlow, M., Osterhaus, A. D., Meineke, R., & Rimmelzwaan, G. F. (2023). Host responses to respiratory syncytial virus infection. Viruses, 15(10), 1999.

  1. Line 230. The sentence: In addition, "human monocytes with the CX3CR1 M280 variant are known to be deficient in CX3CL1- 231 mediated survival" is unclear: Please rephrase and explain.

Dear reviewer, thank you for your commentary. We re-phrased the paragraph in order to clarify it.

Paragraph with corrections:

The CX3CR1 M280 variant has been associated with decreased adhesive capacity of human monocytes [53], hence patients homozygous for CX3CR1 M280 may have decreased transmigration of monocytes and other immune system cells to the infected lung and this may contribute to the severe cases of bronchiolitis reported; and also with a decrease in the ability to rescue death by apoptosis in response to receptor stimulation with CX3CL1 (Collar et al. 18).  In the context of RSV pathophysiology, monocytes are recruited through chemokines secreted by infected epithelial cells in the replicative phase of infection and they are important to limit the infection [54,55]. A decrease in the responses could imply that fewer monocytes migrate into the lung and also, they could exhibit poor survival on lung. Thus, the antiviral activity associated with monocytes would be diminished.

Collar, A.L.; Swamydas, M.; O’Hayre, M.; Sajib, M.S.; Hoffman, K.W.; Singh, S.P.; Mourad, A.; Johnson,
M.D.; Ferre, E.M.; Farber, J.M.; et al. The Homozygous CX3CR1-M280 Mutation Impairs Human Monocyte
Survival. JCI Insight 2018, 3, e95417, 95417, doi:10.1172/jci.insight.95417

Obs1: The authors need to have their manuscript proof-read by a native English speaker. The authors attended this suggestion

Round 2

Reviewer 1 Report

Comments and Suggestions for Authors

The authors have improved the manuscript; however, some points still need to be addressed.

- The authors must include a sentence connecting the viral infection's clinical relevance with the RSV co-receptor information in the abstract.

- In this paragraph: In infants and elderly people, RSV can cause severe lower respiratory tract infections, which can lead to bronchiolitis, pneumonia, and other complications that may result in hospitalization and, in some cases, death [1]. RSV infections in infants can produce long-term pulmonary sequelae (Malinczak, 2020). Although infrequent extrapulmonary manifestations of RSV are very severe and include heart failure, neurological involvement, and hepatitis (Eisenhut, M, 2006), the authors must add more and current references. This is for all newly added references in the text.

- Change elderly for older adults.

- The authors did not include the genotypes of RSV, only the serotypes.

- This: "After a relevant statement, the authors must cite the proper reference", is not addressed in the added information.

- This point needed to be adequately addressed. AEC must look at this cell type. The cell in B must be identified. C needs to be better understood.

Comments on the Quality of English Language

The quality is fine, but the grammar still needs to be corrected.

Author Response

September 1, 2024

Reviewer 1

Corrections for IJMS

Title: The Role of the CX3CR1-CX3CL1 Axis in Respiratory Syncytial Virus Infection and the Triggered Immune Response

Author(s): Selma Rivas-Fuentes*, Alfonso Salgado-Aguayo, Teresa Santos-Mendoza, and Edgar Sevilla-Reyes

ID: ijms-3117624

Dear Reviewer,

The authors appreciate the time you dedicated to reviewing our work, and we are thankful for your valuable comments, which have enhanced it.

We have made the corresponding changes in the text, and we have addressed your observations below.

Comments for reviewer 1, round 2:

            “
The authors have improved the manuscript; however, some points still need to be addressed”

Concerns

  1. The authors must include a sentence connecting the viral infection's clinical relevance with the RSV co-receptor information in the abstract.

Dear reviewer, thank you for your commentary. To address your concern, we have included new information on the abstract, shown with bold lettering.

Abstract: Respiratory syncytial virus (RSV) is a common respiratory pathogen known for causing respiratory illnesses, ranging from mild symptoms to severe lower respiratory tract infections in infants and older adults. This virus is responsible for one-third of pneumonia deaths in the pediatric population, but there are currently only a few effective vaccines. A better understanding of the RSV-host relationship at the molecular level may lead to a more effective management of RSV-related symptoms. The fractalkine (CX3CL1) receptor (CX3CR1) is a co-receptor for RSV expressed by airway epithelial cells and a diversity of immune cells. The RSV G protein binds to the CX3CR1 receptor via a highly conserved amino acid motif (CX3C motif), which is also present in CX3CL1.  The CX3CL1-CX3CR1 axis is involved in the activation and infiltration of immune system cells into the infected lung. The presence of the RSV G protein alters the natural functions of the CX3CR1-CX3CL1 axis, and modifies the host immune response, aspects that need to be considered in the development of an efficient vaccine and specific pharmacological treatment.

Keywords: RSV; CX3CL1; CX3CR1 and Immune Response

  1. In this paragraph: In infants and elderly people, RSV can cause severe lower respiratory tract infections, which can lead to bronchiolitis, pneumonia, and other complications that may result in hospitalization and, in some cases, death [1]. RSV infections in infants can produce long-term pulmonary sequelae (Malinczak, 2020). Although infrequent extrapulmonary manifestations of RSV are very severe and include heart failure, neurological involvement, and hepatitis (Eisenhut, M, 2006), the authors must add more and current references. This is for all newly added references in the text.

According to the reviewer suggestion, specifically in this paragraph we added the following reference: Bottino, 2021; corresponding to the reference 4 in the revised version of the manuscript. Additionally, we have updated the references such that 40 % of all references are from 2019 to date. 

  1. Change elderly for older adults. We have made the requested changes (lines 16, 41 and 52)

  1. The authors did not include the genotypes of RSV, only the serotypes:

We appreciate the reviewer's comment. We have included the number of genotypes once described and frequently referred in the literature, but at the same time referred to papers where they challenged those very same classifications. Furthermore, we have included a reference to more recent work on lineage classification using complete genomes, using a similar framework to that of the PANGO Network for SARS-CoV-2. To address your comments, more data (shown with bold lettering) and two references were added to this paragraph as follows:

RSV is a pleomorphic enveloped virus with three possible morphologies: spherical (ranging in diameter from 100 to 150 nm), asymmetric and filamentous (with filament lengths from 0.5 to 12 μm) [16]. Its genome is a non-segmented, negative-sense, single-stranded RNA of 15.2 kilobases, comprising 10 ORFs, encoding 11 proteins: NS1, NS2, N, P, M, SH, G, F, M2.1, M2.2 and L [17]. RSV has two major subroups: A and B, which often co-circulate, although one subtype usually predominates each season [18], but recent comprehensive work has reclassified RSV diversity into subgroups A and B, and then into lineages based on phylogenetic analyses and amino acid markers of complete genomes RSV has two major antigenic groups, known as subgroups or subtypes: A and B, although one usually predominates each season [23]. Based on the genetic diversity of the viral coded G protein, 13 genotypes have been proposed for RSV-A (Muñoz-Escalante, 2019; Goya, 2020) and 37 for RSV-B (Muñoz-Escalante, 2021) but due to conflicting data, multiple updates and reclassifications have been proposed. More recently, complete genome studies have led to groupings into lineages [27].

References:

Muñoz-Escalante, J. C., Comas-García, A., Bernal-Silva, S., Robles-Espinoza, C. D., Gómez-Leal, G., & Noyola, D. E. (2019). Respiratory syncytial virus A genotype classification based on systematic intergenotypic and intragenotypic sequence analysis. Scientific Reports, 9(1), 20097.

Goya, S., Galiano, M., Nauwelaers, I., Trento, A., Openshaw, P. J., Mistchenko, A. S., ... & Viegas, M. (2020). Toward unified molecular surveillance of RSV: A proposal for genotype definition. Influenza and Other Respiratory Viruses, 14(3), 274-285.

Muñoz-Escalante, J. C., Comas-García, A., Bernal-Silva, S., & Noyola, D. E. (2021). Respiratory syncytial virus B sequence analysis reveals a novel early genotype. Scientific reports, 11(1), 3452.

After a relevant statement, the authors must cite the proper reference.

We also revised the document and added the corresponding references in the statements that were still missing

References:

Arola, M., Ruuskanen, O., Ziegler, T., Mertsola, J., Näntö-Salonen, K., Putto-Laurila, A., ... & Halonen, P. (1990). Clinical role of respiratory virus infection in acute otitis media. Pediatrics, 86(6), 848-855 (line 40)

Belongia, E. A., King, J. P., Kieke, B. A., Pluta, J., Al-Hilli, A., Meece, J. K., & Shinde, V. (2018, December). Clinical features, severity, and incidence of RSV illness during 12 consecutive seasons in a community cohort of adults≥ 60 years old. In Open forum infectious diseases (Vol. 5, No. 12, p. ofy316). US: Oxford University Press. (line 41)

Moyes, J., Walaza, S., Pretorius, M., Groome, M., von Gottberg, A., Wolter, N., ... & Group, S. (2017). Respiratory syncytial virus in adults with severe acute respiratory illness in a high HIV prevalence setting. Journal of Infection, 75(4), 346-355. (line 41)

Cadena‐Cruz, C., Villarreal Camacho, J. L., De Ávila‐Arias, M., Hurtado‐Gomez, L., Rodriguez, A., & San‐Juan‐Vergara, H. (2023). Respiratory syncytial virus entry mechanism in host cells: A general overview. Molecular Microbiology, 120(3), 341-350. (line 81)

Shahriari, S., Gordon, J., & Ghildyal, R. (2016). Host cytoskeleton in respiratory syncytial virus assembly and budding. Virology journal, 13, 1-11. (line 90)

Garton, K. J., Gough, P. J., Blobel, C. P., Murphy, G., Greaves, D. R., Dempsey, P. J., & Raines, E. W. (2001). Tumor necrosis factor-α-converting enzyme (ADAM17) mediates the cleavage and shedding of fractalkine (CX3CL1). Journal of Biological Chemistry, 276(41), 37993-38001. (line 122) 

Hundhausen, C., Misztela, D., Berkhout, T. A., Broadway, N., Saftig, P., Reiss, K., ... & Ludwig, A. (2003). The disintegrin-like metalloproteinase ADAM10 is involved in constitutive cleavage of CX3CL1 (fractalkine) and regulates CX3CL1-mediated cell-cell adhesion. Blood, 102(4), 1186-1195. (line 122) 

Kehrl, J. H. (2006). Chemoattractant receptor signaling and the control of lymphocyte migration. Immunologic research, 34, 211-227. (line 120 y 128) 

Johnson, S. M., McNally, B. A., Ioannidis, I., Flano, E., Teng, M. N., Oomens, A. G., ... & Peeples, M. E. (2015). Respiratory syncytial virus uses CX3CR1 as a receptor on primary human airway epithelial cultures. PLoS pathogens, 11(12), e1005318. (line 136) 

Das, S., Raundhal, M., Chen, J., Oriss, T. B., Huff, R., Williams, J. V., ... & Ray, P. (2017). Respiratory syncytial virus infection of newborn CX3CR1-deficient mice induces a pathogenic pulmonary innate immune response. JCI insight, 2(17). (line 152)

Zhivaki, D., Lemoine, S., Lim, A., Morva, A., Vidalain, P. O., Schandene, L., ... & Lo-Man, R. (2017). Respiratory syncytial virus infects regulatory B cells in human neonates via chemokine receptor CX3CR1 and promotes lung disease severity. Immunity, 46(2), 301-314. (lines 299 and 301)

According to the reviewer suggestion, we have included the corresponding references in lines: 40, 41, 81, 90, 119, 121, 127, 136, 152, 169, 173, 178, 299 and 301

  1. This point needed to be adequately addressed. AEC must look at this cell type. The cell in B must be identified. C needs to be better understood.

Dear reviewer, thank you for your comment, which was very useful to improve our figure.

In panel A, we replaced the drawing of the AEC cell with a drawing more representative of this cell type, e.g., the elongated apical shape and its cilia. In the figure legend it was specified that CX3CR1+ immune cells can be monocytes, macrophages, T cells, NK cells, Dendritic cells and B cells.

In panel C, for clarity, we depicted how the RSV G protein can function as an antigenic decoy, binding to CX3CR1 receptors, and displacing the natural ligand CX3CL1. In addition, a new figure legend was drafted

New figure legend.

Figure 1. The presence of RSV G protein alters the natural functions of the host immune response. The G protein of RSV binds to the CX3CR1 receptor expressed on the membrane of target cells, such as airway epithelial cells (AEC), and enables infection. This binding occurs through the CX3C domain of the G protein, which mimics the CX3C motif of the chemokine CX3CL1. The complexity of the system is increased because cellular responses to membrane-bound or soluble CX3CL1 binding are also different, and RSV G protein exists in bound and secreted forms (A). Various CX3CR1+ immune cells, such as monocytes, dendritic cells, NK cells, and some subtypes of T and B cells, contribute to the immune response against RSV.  These cells can be infected with RSV and contribute to viral evasion mechanisms (B).  RSV G protein functions as an antigenic decoy and shifts the binding of CX3CL1 to CX3CR1 (C).

  1. The quality is fine, but the grammar still needs to be corrected.

Dear reviewer, thank you for your comments. We received the help of a native english speaker to review the grammar, and we believe that the resulting text improved in aspects such as readability, grammar and clarity. We hope this will remedy the grammar deficiencies you pointed out to us.

Dear reviewer, thank you for helping us to improve our work

Dr. Selma Rivas Fuentes

Reviewer 3 Report

Comments and Suggestions for Authors

The author have adequately addressed the comments.

Round 3

Reviewer 1 Report

Comments and Suggestions for Authors

The authors have appropriately addressed my suggestions.

Comments on the Quality of English Language

The quality is fine.